# Comparison of Femoral Component Rotation between Robotic-Assisted vs. Soft-Tissue Tensor Total Knee Arthroplasty with Anatomic Implants

**DOI:** 10.3390/medicina59050880

**Published:** 2023-05-04

**Authors:** Bartosz M. Maciąg, Tomasz Kordyaczny, Grzegorz J. Maciąg, Marcin Łapiński, Dawid Jegierski, Jakub Świderek, Hanna Tsitko, Monika Dorocińska, Krystian Żarnovsky, Maciej Świercz, Piotr Stępiński, Olga Adamska, Artur Stolarczyk

**Affiliations:** 1Department of Orthopedics and Rehabilitation, Międzyleski Specialist Hospital, Medical University of Warsaw, 04-749 Warsaw, Masovian Voivodeship, Poland; 2Faculty of Medicine, Medical University of Białystok, 15-089 Białystok, Podlaskie Voivodeship, Poland; 3Faculty of Medicine, Medical University of Lublin, 20-059 Lublin, Lubelskie Voivodeship, Poland

**Keywords:** total knee replacement, rotation, Fuzion balancer, RATKA, computed tomography, osteoarthritis, soft tissue tensioner, NAVIO, measured resection technique, conventional total knee replacement

## Abstract

*Background and Objectives*: Total knee arthroplasty (TKA) is the most effective treatment method for end-stage osteoarthritis. One of the most important aspects of this surgery is adequate implant positioning, as it guarantees the desired outcome of restoring limb biomechanics. Surgical technique is being continuously improved along with hardware development. There are two novel devices designed to help establish proper femoral component rotation: soft-tissue tensor and robotic–assisted TKA (RATKA). This study compared the femoral component rotation achieved with the use of three methods: RATKA, soft tissue tensioner and the conventional measured-resection technique, all of them utilizing anatomical design prosthesis components. *Materials and Methods*: A total of 139 patients diagnosed with end-stage osteoarthritis underwent total knee arthroplasty between December 2020 and June 2021. After the surgery, they were divided into three groups depending on procedure technique and implant type: Persona (Zimmer Biomet) + Fuzion Balancer, RATKA + Journey II BCS or conventional TKA + Persona/Journey. Postoperatively, a computed tomography examination was performed in order to measure femoral component rotation. All three groups were compared independently during statistical analysis. Fisher’s exact, Kruskal–Wallis and Dwass–Steel–Crichtlow–Fligner tests were used for particular calculations. *Results*: Statistically significant differences in femoral component rotation between groups were noticed. However, in terms of values other than 0° in external rotation, no significant variance was revealed. *Conclusions*: Additional total knee arthroplasty instruments seem to improve the outcomes of the surgery, providing better component positioning than in the conventional measured-resection technique based only on bone landmarks.

## 1. Introduction

Osteoarthritis (OA) is a complex disease with highly heterogenous characteristics in terms of its features at both the clinical and cellular level. Its course differs significantly between patients and so do patients’ responses to particular treatment methods. At early stages, an attempt can be made to slow down the course of the disease with disease-modifying osteoarthritis drugs (DMOADs) and techniques involving stem cells and their components. There are also symptomatic slow-acting drugs for OA (SYSADOAs); however, their effectiveness has not been proved. Surgical treatment of early-stage OA includes arthroscopy, aiming to debride the joint and stimulate the bone marrow. Another method is osteotomy which is assigned for unicompartmental OA and consists of realigning the joint and thus changing the distribution of weight, alleviating the overstrained compartment. In end-stage OA, total knee arthroplasty (TKA) is considered as the most effective method. Over the years, it has become the gold standard of treatment for advanced OA and, along with total hip arthroplasty, is one of the most commonly performed orthopedic surgeries [1,2,3,4,5].

Despite the growing number of alloplasty procedures performed worldwide, about 20% of patients remain unsatisfied with the results of TKA and about 7% with the result of THA as reported by Anakwe et al. The most common complaints from patients who have received TKA are persistent pain, restricted range of motion (ROM), stiffness, or the feeling of an “unnatural” knee. Another important difference concerns the functional outcome of alloplasty—it was noted that the number of physically active post-TKA patients decreases, while in THA it increases [6,7,8].

It is widely believed that proper implant positioning is essential in increasing the likelihood of a successful outcome, as it plays an important role in avoiding soft tissue impingement and the loosening of the prosthesis. It is also crucial in restoring native knee biomechanics, by allowing the knee to move in a full ROM with proper stability [9,10].

In recent years, utmost attention has been paid to the proper rotational alignment of the femoral component. External rotation occurs when the sagittal axis of the component deviates outwards, while internal rotation occurs when the component’s axis deviates inwards. In the study by Newman et al., where authors analyzed 190 computed tomography (CT) images, it was stated that the native femur is internally rotated within 3° in 65% of healthy knee joints. It has been proven that along with the progression of osteoarthritis, this rotation might increase due to increased soft tissue tension resulting from increasing lower limb malalignment. In order to obtain an even flexion gap during TKA, the femoral component ought to be rotated in the opposite direction with the same degree value. Kim et al. proved that a femoral component external rotation between 2° and 5° provides a lower rate of implant failure postoperatively. It has also been proven that excessive external femoral component rotation leads to postoperative flexion and midflexion instability, pain, stiffness and anterior knee pain. On the other hand, internal rotation may lead to patellar maltracking which results in patellar instability [11,12,13,14,15,16].

In the recent consensus study by the European Knee Associates—European Society of Sports Traumatology, Knee Surgery and Arthroscopy (EKA—ESSKA), it was stated that there is no perfect technique to establish proper femoral component rotation. The most popular bone landmarks to perform measured-resection (MR) technique—transepicondylar axis (TEA), posterior condylar axis (PCA) and anteroposterior axis (APA)—have no superiority over each other and thus surgeons should take all of them into account while performing TKA. In recent years, another technique has been developed—gap balancing (GB). This technique utilizes soft tissue tension assessment to establish a level of posterior condyles resection sufficient to obtain an even, balanced flexion gap. In recent systematic reviews and meta-analyses, no superiority of either technique was demonstrated, as both have some significant limitations. In the MR technique, a poor soft tissue envelope may lead to the failure of the procedure despite perfect bone cuts. On the other hand, a GB varus or valgus proximal tibial cut can possibly lead to internal or excessive external rotation of the femoral component [15,17,18,19,20,21,22].

In recent years, several devices expected to facilitate TKA have been brought to market, such as robots or dynamic tensioners. It is believed that their accuracy will lower the incidence of femoral component malrotation and will thus improve TKR outcomes. It is important to note that, to date, no authors have proven the superiority of either robotic-assisted (RATKA) or dynamic-tensioner-utilization total knee arthroplasty [23,24,25].

The aim of this study was to compare the values of the femoral component rotation between three different TKA methods: RATKA, TKA with dynamic tensioner—Fuzion, and conventional TKA without any additional instrumentation. Each method utilized anatomical shaped prosthesis components.

## 2. Materials and Methods

This study was conducted according to the STROBE (strengthening the reporting of observational studies in epidemiology) statement and an appropriate checklist was presented to the editors. Informed consent was obtained from all subjects involved in this study.

Patients included in the study were >50 years of age; had clinically and radiologically confirmed moderate-to-severe osteoarthritis (osteophytes, narrowed joint space, subchondral osteolysis); were treated with standard doses of non-steroid anti-inflammatory drugs for a minimum of three years before surgery and were undergoing primary TKA with a posterior-stabilized (PS) implant without patellar resurfacing; had at least a 15°-flexion contracture and had a post-TKA CT scan available for assessment. Exclusion criteria included (i) patients with prior high tibial osteotomy (HTO) or other lower limb surgery; (ii) patients with rheumatoid arthritis or positive rheumatoid factor in serum; (iii) patients with an erythrocyte-sedimentation rate or C-reactive protein over the normal range; (iv) patients who qualified for cruciate-retaining implants (posterior cruciate ligament—PCL—intact at the time of surgery and absent flexion contracture preoperatively). Inclusion and exclusion criteria are both shown in Table 1 [26].

A consecutive series of patients qualified and were operated on by two fellowship-trained surgeons at the level III academic hospital between December 2020 and June 2021. They were operated on with the on-label use of PERSONA PS (Zimmer-Biomet, Warsaw, IN, USA) or Journey II BCS (Smith and Nephew, Watford, UK) implants without patella resurfacing as a treatment for end-stage knee osteoarthritis. Patients undergoing PERSONA PS TKA were operated on with the use of the dynamic balancer Fuzion (Fuzion group) and patients operated on with Journey II BCS were operated on with the use of the RATKA system Navio (Navio group). Two cohorts were then compared with a matched-cohort of patients who underwent conventional TKA with the use of either PERSONA PS or Journey II BCS (conventional measured-resection technique group, later referred to as the conventional group) without any additional devices. This group consisted of subjects with both PERSONA PS, as well as Journey II BCS prosthesis, implanted in order to minimize the implant-related risk of bias. For both cohorts, a propensity score was generated based on gathered demographic patient data (age at surgery, sex, and body mass index—BMI). A summary of the demographic data is depicted in Table 2. Fuzion and Navio patients were matched to patients from the conventional group using a 0.1 propensity score threshold with priority given to exact matches.

All surgeries were performed with use of a tourniquet (average time of 65 min) with postoperative closed suction drainage. All surgeries were performed using a standard midline incision and medial parapatellar arthrotomy. Cruciate-sacrificing implants were used, and tibial cuts were performed first using extramedullary alignment jigs. These cuts were made perpendicular to the long axis of the tibia with a posterior slope between 0° and 7° according to the native value. Then, a distal femoral cut was performed, and the extension gap balance was assessed. The femur was prepared using an intramedullary alignment tool with a valgus correction angle between 5° and 7°. The external rotation of the femoral component was assessed using three different techniques.

The first group received surgery aided by the use of the dynamic balancer Fuzion aiming to achieve symmetrical 2-mm opening gaps on both the lateral and medial sides in 90° of flexion after osteophytes removal as in technique described by Benazzo et al. [27].

The second group received surgery aided by the use of the robotic-assisted system NAVIO with targeted symmetrical 2-mm opening gaps on both the lateral and medial sides in 90° of flexion after osteophytes removal [28].

The third group received surgery aided by the standard measured-resection technique consisting of flexing the knee and evaluating the relation of the posterior condylar axis with the transepicondylar and Whiteside’s line. The initial bone cuts were performed independent of soft tissue tension. The femoral implant size was picked with the use of the anterior referencing technique [29].

Femoral bone cuts were made in the sequence recommended by the surgical protocol of the PERSONA knee system and Journey II BCS system. Flexion and extension gaps were balanced. No patellar resurfacing was performed. All components were implanted with the use of cement. The postoperative protocol included chemical and mechanical thromboprophylaxis unless specifically contraindicated. All patients received one dose of parenteral antibiotics at the induction of anesthesia and two further doses postoperatively.

During the first 24 h after the surgery, a computed tomography examination of the lower limb was performed by a skilled musculoskeletal radiologist (Philips Incisive CT; Philips Healthcare, Cleveland, OH, USA).

For the measurement of the femoral component rotation, the method proposed by Berger et al. was used. On the CT horizontal scan, two lines were drawn where the lateral and medial epicondyles of the femur were the most prominent: a line between the surgical TEA (connecting the lateral epicondyle and medial sulcus of the medial epicondyle) and a line connecting the medial and lateral prosthetic posterior condylar surface (posterior condylar axis—PCA). The angle between these two lines form the angle of component rotation. Depending on the side where the vertex of the angle is located—on the lateral or medial side—rotation is determined as external or internal, respectively [29].

Analysis of the radiographic images was performed using the INFINITT PACS system (Infinitt Healthcare, Seoul, Republic of Korea).

All scans were measured three times by two independent researchers, and mean values of their results were noted. In order to avoid the potential risk of bias, all data concerning participants were blinded. Mean intra- and interobserver differences in measurements of femoral and tibial components were calculated for all cases. Intra- and interobserver reliability were determined by calculating the intraclass correlation coefficient with a confidence interval (CI) of 95%. The method of measuring the rotation angle is shown in Figure 1.

Statistical analysis of the results was performed by a statistician. Since all comparisons regarded three groups, Fisher’s exact test was used to analyze categorical variables and Kruskal–Wallis test was used to compare continuous variables with a post-hoc pairwise analysis using Fisher’s exact test with Bonferroni correction and Dwass–Steel–Crichtlow–Fligner test (DSCF), accordingly. All comparisons were performed between independent groups. An α value of 0.05 was used to determine the statistical significance of all the analyses.

## 3. Results

A total of 31 patients who were treated with the use of Fuzion, 61 with the use of Navio and 39 without using any additional devices met the inclusion criteria.

There were statistically significant differences in femoral component rotation between the groups (*p* = 0.0137). Post-hoc analysis showed a significant difference between the Fuzion and Navio groups (2.42 ± 2.87 vs. 4.05 ± 1.89, *p* = 0.0321) and an almost significant difference between the measured-resection and Navio groups (3.46 ± 1.86 vs. 4.05 ± 1.89, *p* = 0.0613), where a positive value indicates external rotation. The difference between the Fuzion and measured-resection groups was not statistically significant (*p* = 0.78).

An analysis of the values of rotation above the value of 0° of external rotation showed significant differences between the groups (*p* = 0.0124). However, a post-hoc pairwise comparison showed no statistically significant difference between any two groups. An analysis of the femoral rotation values different from 3° of external rotation showed no significant differences between the groups (*p* = 0.14). All the results of the statistical analysis are depicted in Table 3, Table 4, Table 5, Table 6 and Table 7. Table 3 and Figure 2 present the mean femoral component rotation values achieved with the use of each technique, along with the significance of differences between particular techniques expressed as *p*-values. Table 4 presents the number of examined subjects with external and 0° rotation of the component over the total number of subjects in the group of each used technique and with the significance of differences between particular techniques expressed as *p*-values. Table 5 depicts the number of subjects with internal rotation of the component over the total number of subjects in the group and with the significance of differences between particular techniques expressed as *p*-values. Table 6 presents the number of subjects with 3° of external rotation over the total number of subjects in the group and with the significance of differences between particular techniques expressed as *p*-values. Table 7 presents the range of rotation values in each group, where a positive value indicates external rotation of the component, along with its statistical significance.

## 4. Discussion

One of the most important surgical goals of TKA is restoring the stability of the knee in a full range of motion with proper patellar tracking, by achieving proper rotational alignment of the implant in all dimensions. Due to the persistent percentage of patients not fully satisfied with TKA, there has been an ongoing debate to establish the gold standard of surgical technique, with hopes of achieving the goal of the “forgotten knee joint”—when during everyday activities, the patient forgets about the fact that the knee was replaced by an artificial one. This can be only achieved when the TKA procedure is performed perfectly.

In recent years, the anatomic design of total knee components has been introduced to the market. With the higher availability of femoral sizes, oblique joint line inserts and asymmetrical tibial baseplates, these designs were believed to be the remedy for unsatisfied patients, as they are expected to improve the implant fit. In in vitro studies, it has been proven that these designs, such as PERSONA or Journey II BCS, may lower the risk of overhang on both femur and tibia and therefore decrease the risk of pain following TKA. With their precise instruments, both of these designs allow femoral rotation to be set to different values from the most commonly used 3° of external rotation [30].

To the authors’ best knowledge, thus far, there is a lack of studies analyzing values of femoral rotation following anatomic design TKA with the use of dedicated rotation assessment instruments such as a robotic-assisted system or dynamic tensioner.

Even though in recent systematic reviews RATKA did not show superiority in terms of clinical outcome among experienced surgeons, there is still a lack of studies reporting the radiological outcomes of this procedure. The Navio system utilized in this study is a semi-automatic system that follows the reamer’s trajectory. When a surgeon deviates from a planned bone cut, the reamer stops, which prevents them from making imprecise resection. The robot assesses the femoral rotation by analyzing bone marks, in the same way as in the MR technique. The difference is that it allows for a very precise adjustment of this value for a perfectly balanced flexion gap, equally opening in both compartments for 2 mm. In fact, the technique of femoral rotation assessment for NAVIO might be described as a hybrid technique between MR and GB. Despite their accuracy, RATKA systems are also susceptible to surgeons’ mistakes—the precision of the whole navigation system is granted by the proper placement of bone markers by the surgeon.

Another big advantage of the Navio system is that it does not require preoperative CT scans because it relies only on intraoperative data.

Fuzion is one of the dynamic tension balancers utilized in primary TKA. It allows surgeons to assess soft tissue tension before any bone cuts are made in order to obtain an equally balanced gap in 90° of flexion and extension intraoperatively. In a study by Benazzo et al., no statistically significant superiority of Fuzion over standard methods was proven; however, importantly in their study, TKR with the use of a Fuzion tensioner had a higher deviation of external rotation of the femoral component. The authors stated that this technique might be the key to optimizing the outcomes and achieving the state of the “forgotten knee”. These results correspond partially with the results of our study, as the use of Fuzion allowed us to implant femoral components in internal rotation and in higher values of external rotation, even though the average value of rotation was slightly below 3° [27].

Apart from strictly surgical aspects, economic factors should also be considered. In the case of the dynamic balancer Fuzion, the device itself is a part of the surgical equipment provided by the company. Where robotic navigation systems are concerned, not only does the hospital have to carry the cost of the purchase of the navigation system but also the cost of every single surgery is higher due to the single-use instruments employed. Most hospitals struggle with a deficiency of financial resources; thus, Fuzion may occur as a more available option and could be easier to implement on a larger scale.

Time of surgery is yet another matter that differs between the two methods. RATKA surgeries take more time than those with the use of a dynamic tensioner, and thus the exposure time of the wound is longer. It is a well proven fact that the longer the surgery, the higher the risk of wound infection which in cases of periprosthetic infections (PJI) is extremely important to consider due to significant difficulties in PJI treatment [31].

In terms of implicating these two novel devices, another important aspect one has to consider is the learning curve. Since the dynamic tensioner is a strictly mechanical instrument, it is easier to master. In the robotic navigation system, the surgeons must deal not only with the new hardware but also the need to master operating the software. This may prove especially troublesome among older doctors.

Another controversial topic is the alignment concept in TKA. The standard mechanical alignment concept gains more and more critics. On the contrary, the concept of restricted kinematic alignment is becoming more popular. A crucial aspect of this technique is a proper understanding of the initial limb deformity and its precise correction. This is the area where robotic navigation systems gain an advantage over tensioners which cannot measure the value of mechanical axis deviation.

In the studies by Sharma et al. and Figueroa et al., the authors confirmed that CT is the optimal way to assess the rotational alignment of knee components with a low risk of inter- and intraobserver unreliability [32,33].

In the study by Heesterbeek et al., the authors concluded that performing internal rotation of the femoral component might allow for achieving a balanced flexion gap. Although there is some evidence that such rotation might have some negative influence on patellofemoral tracking and knee biomechanics, there is no strong evidence that the pain corresponding to the patellofemoral joint is caused by a malrotated femoral component. In certain cases, it is acceptable to leave the component internally rotated in order to obtain a balanced flexion gap, but some surgeons would rather accept medial laxity in flexion to avoid internal rotation. However, according to Tsukiyama et al., medial incongruency in the knee joint after knee arthroplasty is not as well tolerated as lateral, which is a strong suggestion towards reconsideration of the aforementioned practice [34,35].

It is also important to notice the limitations of our study: there is a lack of clinical outcome measurement, so the influence of femoral rotation on patients’ satisfaction is unknown; CT scans allow only for a static evaluation of components’ positioning and do not allow a full understanding of their relations in motion. Even though both PERSONA and Journey knee implants are considered anatomic ones, it is important to note that additional instruments for precise assessment of femoral rotation are compatible only with particular prosthesis designs, so a comparison of Fuzion and NAVIO could not be performed on both designs. Furthermore, this is a matched-cohort study only, without any preoperative randomization process.

## 5. Conclusions

Recently provided anatomically shaped designs of TKA implants are thought to be the next step in improving the outcomes of TKA surgery. Additional instruments that might be used intraoperatively allow surgeons to establish femoral component rotation in a more precise way than by using anatomic bone landmarks only. Further studies must be performed on larger cohorts of patients, including randomized-controlled studies, assessing the role of RATKA and dynamic tensioners in establishing desired femoral component rotation in TKA with the use of anatomic knee designs and its impact on the functional outcome.

## Figures and Tables

**Figure 1 medicina-59-00880-f001:**
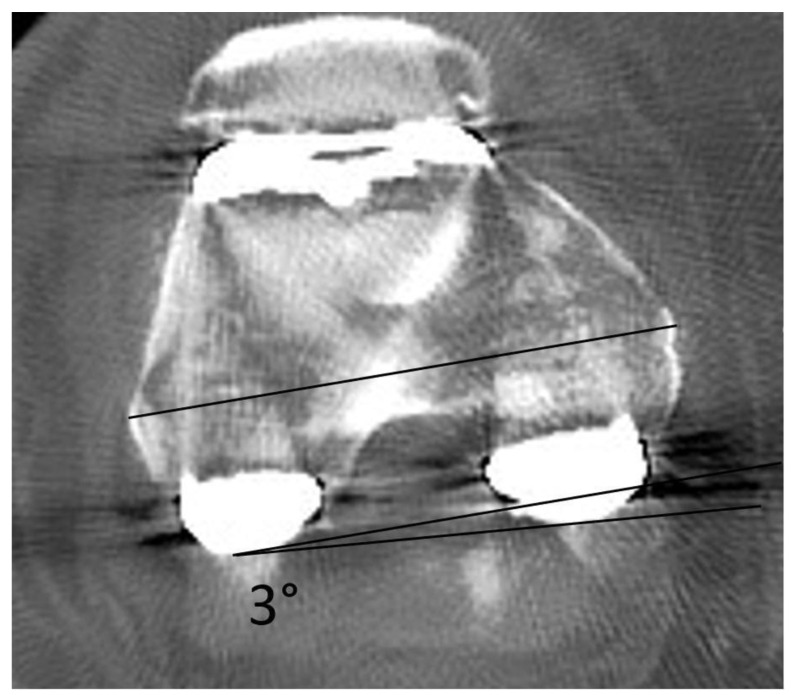
Method of measurement of the femoral component rotational alignment—in this case, 3° of internal rotation.

**Figure 2 medicina-59-00880-f002:**
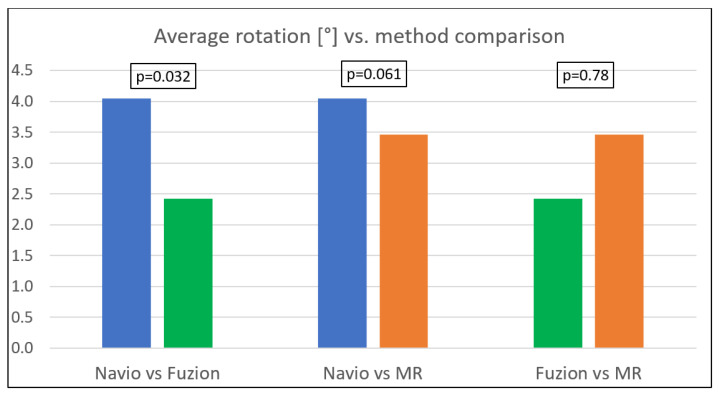
Average rotation of the femoral component comparison between groups. Statistical significance only in the first column.

**Table 1 medicina-59-00880-t001:** Inclusion and exclusion criteria.

Inclusion Criteria	Exclusion Criteria
Age > 50	Prior HTO or another lower limb surgery before TKR
Confirmed osteoarthritis treated with standard doses of non-steroid anti-inflammatory drugs before surgery	Rheumatoid arthritis, positive rheumatoid factor in serum
Moderate-to-severe osteoarthritis lasting more than three years	Erythrocyte-sedimentation rate or C-reactive protein over the normal range
Primary TKA, PS implant, no patella resurfacing	Qualification for a CR implant
Flexion contracture of the knee >15°	
Post-TKA CT scan available for assessment	

**Table 2 medicina-59-00880-t002:** Baseline characteristics of the patients.

	Baseline Group Characteristics			
	Fuzion	Measured Resection Technique	Navio	*p*-Value
BMI (body mass index—kg/m^2^)	30.15 ± 4.01	30.28 ± 4.42	31.29 ± 4.93	0.46
Age (years)	68.81 ± 7.31	71.16 ± 7.13	67.25 ± 6.05	<0.05
Male: female (No. of subjects)	12:19	18:21	15:46	0.0678

**Table 3 medicina-59-00880-t003:** Comparison of obtained mean femoral component rotation between examined groups and its statistical significance.

		Comparison between Groups (*p*-Value)		
Group	Mean Rotation (Degrees); Positive Value Indicates External Rotation	vs. Fuzion	vs. Measured Resection Technique	vs. Navio
**Fuzion**	2.42 ± 2.87	-	0.78	**0.0321**
**Measured-resection technique**	3.46 ± 1.86	0.78	-	0.0613
**Navio**	4.05 ± 1.89	**0.0321**	0.0613	-
**Difference between the groups**		**0.0137**		

**Table 4 medicina-59-00880-t004:** Comparison of obtained external femoral component rotation (angle ≥ 0°).

			Comparison between Groups (*p*-Value)		
Group	Number of Subjects (Post-TKA Subjects with External Rotation of the Component/Total Number of Subjects in Group)	Percentage	vs. Fuzion	vs. Measured Resection Technique	vs. Navio
**Fuzion**	26/31	83.87	-	>0.05	>0.05
**Measured-resection technique**	39/39	100.00	>0.05	-	>0.05
**Navio**	59/61	96.72	>0.05	>0.05	-
**Difference between the groups**			**0.0124**		

**Table 5 medicina-59-00880-t005:** Comparison of obtained internal femoral component rotation (angle < 0°) between groups.

			Comparison between Groups (*p*-Value):		
Group	Number of Subjects (Post-TKA Subjects with 0° and Internal Rotation of the Component/Total Number of Subjects in Group)	Percentage	vs. Fuzion	vs. Measured Resection Technique	vs. Navio
**Fuzion**	5/31	16.13	-	>0.05	>0.05
**Measured-resection technique**	0/39	0.00	>0.05	-	>0.05
**Navio**	2/61	3.28	>0.05	>0.05	-
**Difference between the groups**			**0.0124**		

**Table 6 medicina-59-00880-t006:** Comparison of obtained correct femoral component rotation (angle = 3°) between groups.

			Comparison between Groups (*p*-Value):		
Group	Number of Subjects	Percentage	vs. Fuzion	vs. Measured Resection Technique	vs. Navio
**Fuzion**	8/31	25.81	-	>0.05	>0.05
**Measured-resection technique**	16/39	41.03	>0.05	-	>0.05
**Navio**	14/61	22.95	>0.05	>0.05	-
**Difference between the groups**			0.14		

**Table 7 medicina-59-00880-t007:** Range of the femoral component rotation angles in each group. Positive value indicates external rotation.

Group	Number of Subjects	Minimum (Degrees)	Maximum (Degrees)
Fuzion	31	−3.00	7.00
Measured-resection technique	39	0	8.00
Navio	61	−2.00	9.00

## Data Availability

The datasets used and/or analyzed during the current study will be available from the corresponding author on reasonable request.

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
