# Peer review of "Comparison of Femoral Component Rotation between Robotic-Assisted vs. Soft-Tissue Tensor Total Knee Arthroplasty with Anatomic Implants"

_medicina, 2023, doi:10.3390/medicina59050880_

Round 1
Reviewer 1 Report
Medicina
Manuscript ID
medicina-2264882
Title: Comparison of femoral component rotation between robotic- assisted vs utilizing soft tissue tensor total knee arthroplasty with anatomic implants.
Comments:
Abstract: Reference must be removed from abstract. Objective is weak. It needs sufficient clarification
Introduction:
1.Total knee arthroplasty is considered as gold standard of treatment for end-stage knee osteoarthritis. Why ? What are the other methods available? Please provide information along with references.
2. What is THA treatment and how is it different from TKA. Which treatment is more acceptable. Incorporate information along with references
3. Information regarding full ROM movement with proper stability, patellar instability etc must be incorporated.
4. Limitations of gap balancing must be incorporated
5. Reference must be incorporated for the line “However, in several reviews ….. tensioners was observed”.
6. Line “It is believed that thanks to their …..reported out comes”, not appropriately. Unable to draw correct information.
7. What is femoral component rotation and how it is important in the study. Information along with reference must be incorporated.
8. Internal, external rotation etc must be defined. Advantage/disadvantage must also be mentioned along with references.
9. Objectives needed to be justified with sufficient clarification.
Methods
1. How subjects were considered in the study and what are the clinical criteria( like age, sex, duration of study, whether severe/non severe, medicine recommended, ESR, CRP, RF etc.), must be provided in table format.
2. Parameters considered under radiologic examination must be incorporated
3. In first group 90 degrees of flexion after osteophytes removal is mentioned whereas in second group it is mentioned that both lateral and medial sides in 90 degrees of flexion. However in 3rd group there is no information about 90 degrees of flexion. Why? Please clarify the importance of “90 degrees of flexion” in each group.
4. What are the strategy of making different groups of patients? How many group of patients were made and in each group how many patients were involved.
5. Berger Method must be incorporated in brief.
6. How analysis of the radiographic images was performed using the INFINITT PACS system. Method must be incorporated in brief along with reference.
7. Inclusion criteria of patients must be included.
8. Criteria for qualification of patients for cruciate-retaining implants must be incorporated along with references. Incorporate information if PCL intact at time of surgery is beneficial, how does it create value to the patients, how many patients were qualified and whether any kind of specific diet enhance the beneficial value. Please incorporate information.
9. Information about PERSONA PS, Journey II BCS, must be incorporated along with references. Why this 2 methods were attempted.
Results:
1. Why femoral component rotational alignment was 3° of internal rotation in the said study. Was there any optimization study carried out. Justify along with relevant information and references.
2. Analysis of all the results is missing. It is hard to understand the relevant data shown in each table. Result obtained in each table must be analysed and incorporated.
3. The results obtained in all the table must be explained.
Figs/table
1. Table 1 is missing.
4. Table legend is not sufficiently informative and must be placed appropriately
5. In table 2 and 3, value shown in number of subjects column is not clear. What does the value 26/31, 39/39, 59/61……etc indicate.
6. Full form of all short form used must be incorporated as a foot note.
Discussion:
1.What is NAVIO, FUZION. How it is relevant to OA. Information must be incorporated.
2. Why CT is the considered as gold standard to assess rotational alignment. Comparison with other available method and advantage of CT method must be incorporated. Relevant discussion along with reference must be incorporated.
3. What is dynamic balancer FUZION and RATKA system and what does Navio group represents. Information must be incorporate
Conclusion:
The study has not been well organized and not well written. The objective is not sufficiently strong. The importance of each method attempted and methodology has not been well elaborated. Also the data obtained is not properly analysed and concluded meaningfully. There are many gaps, hence confusion.
Others
Full form of all Abbreviation (TKA,TKR, THA, ROM, HTO, NAVIO, FUZION,CT, PCL) used must be incorporated when appear 1st time in the manuscript.

Reviewer 2 Report
This is an interesting and well designed study. there are few points which the authors can address in the revision
1) details of the measurement can be explained in more details. This will help in making the data easily replicable
2) The tables can be simplified for easy comprehension. The authors can attempt to represent the data in a graph with p values.
3) The authors can discuss about the potential application and limitation of the study
4) The authors can compare the literature results with current data.
Round 2
Reviewer 1 Report
Revised comments
1) The line number must be mentioned in which the response of comments incorporated in the revised manuscript
2)In Method section, response to comment 1 is not satisfactory. The information as per the comments raised (duration of disease, whether severe/non severe, medicine recommended, ESR, CRP, RF) must be incorporated.

Author Response
|
1) The line number must be mentioned in which the response of comments incorporated in the revised manuscript |
We mention the lines number in our responses. |
|
2)In Method section, response to comment 1 is not satisfactory. The information as per the comments raised (duration of disease, whether severe/non severe, medicine recommended, ESR, CRP, RF) must be incorporated. |
We completed the table 1 with proper information.
|
Reviewer 2 Report
the authors have addressed the reviewers comment and the manuscript can be accepted after necessary editorial changes
Author Response
the authors have addressed the reviewers comment and the manuscript can be accepted after necessary editorial changes